# Association of Indolebutyric Acid with *Azospirillum brasilense* in the Rooting of Herbaceous Blueberry Cuttings

**Renata Koyama, Walter Aparecido Ribeiro Júnior, Douglas Mariani Zeffa[ID], Ricardo Tadeu Faria, Henrique Mitsuharu Saito, Leandro Simões Azeredo Gonçalves[ID] and Sergio Ruffo Roberto *[ID]**

Agricultural Research Center, Londrina State University, Celso Garcia Cid Road, km 380, P.O. Box 10.011, Londrina ZIP 86057-970, Brazil; emykoyama@hotmail.com (R.K.); junior_agro40@hotmail.com (W.A.R.J.); douglas.mz@hotmail.com (D.M.Z.); faria@uel.br (R.T.F.); henriquesaito.m@gmail.com (H.M.S.); leandrosag@uel.br (L.S.A.G.)
* Correspondence: sroberto@uel.br

**Abstract:** Association between auxins and plant growth-promoting bacteria can stimulate root growth and development of fruit crop nursery plants, and can be a promising biological alternative to increase the rooting of cuttings. The objective of this study was to assess the viability of producing 'Powderblue' blueberry nursery plants from cuttings using different doses of indolebutyric acid (IBA) in association with *Azospirillum brasilense*. The following treatments were tested: 0 (control); 500 mg L$^{-1}$ of IBA; 1000 mg L$^{-1}$ of IBA; *A. brasilense*; 500 mg L$^{-1}$ of IBA + *A. brasilense*; and 1000 mg L$^{-1}$ of IBA + *A. brasilense*. The experimental design was completely randomized, with six treatments and four replicates, and each plot (box) consisted of 10 cuttings. The boxes were arranged in a mist chamber with an intermittent regimen controlled by a timer and solenoid valve. After 90 days, the following variables were assessed: rooted cuttings; survival of cuttings; foliar retention; sprouting; cuttings with callus; root dry mass per cutting; number of roots per cutting; and root length. It was observed that the application of IBA with the *A. brasilense* rhizobacteria increased the number of roots of 'Powderblue' blueberry cuttings, while the treatments with IBA alone and IBA 1000 mg L$^{-1}$ + *A. brasilense* increased the root length of cuttings. However, treatments with IBA and *A. brasilense* had no impact on % rooted cuttings and % survival of cuttings.

**Keywords:** auxin; rhizobacteria; rooting; Vaccinium spp.

## 1. Introduction

Blueberry (*Vaccinium* spp.), a fruit species of temperate climates, has high medicinal value owing to its high content of anthocyanins which confer high antioxidant properties to the fruits. These antioxidants help neutralize free radicals related to the onset of degenerative diseases, such as cancer, cataracts, immune disorders, cognitive impairments, and muscle degeneration [1]. The commercial interest in this species has greatly increased because of several studies reporting its efficacy as a medicine [2–4]. Consequently, its cultivation is becoming widespread and of interest to producers in Brazil, where it was first grown around 1983 in Pelotas (RS), in studies conducted by the Brazilian Agricultural Research Corporation [5].

Blueberry is mostly asexually propagated through cuttings as adult plants are obtained in a shorter time period compared to those sexually propagated. In addition, this method can be easily executed and lead to the formation of adventitious roots in cuttings in any season of the year [6,7]. However, different blueberry cultivars show distinct responses regarding rooting of cuttings owing to genetic factors that influence cell differentiation and root formation [8,9].

For plant species that are difficult to root, such as blueberry, auxiliary techniques with growth regulators, such as indolebutyric acid (IBA), can be used [10]. Exogenous application of synthetic auxins is very common in vegetative propagation methods using cuttings as they act similar to the natural phytohormone [11]. However, studies have shown that the use of the synthetic hormone IBA in high concentrations can be toxic to the plant or inhibit rooting [12].

The use of plant growth-promoting rhizobacteria (PGPRs) can be a promising biological alternative for increasing the rooting of cuttings [13]. *Azospirillum* is a genus of PGPRs that inhabits the roots of host plants and provides beneficial effects to the plant under normal growth and/or stress conditions [14]. PGPRs of this genus can increase the fixation of free nitrogen and the production of phytohormones, thus promoting growth in inoculated plants [15].

Studies have shown promising results for association with the PGPR species *Azospirillum brasilense*; it promotes root development by increasing the production of hormones, leading to growth and development of plants [16–18]. In this context, the objective of this study was to assess the viability of producing blueberry nursery plants from cuttings using different doses of IBA in association with *A. brasilense*.

## 2. Materials and Methods

### 2.1. Plant Materials

The experiment was conducted from April to July 2018 at the Department of Horticulture of the Agricultural Research Center of the Londrina State University—PR, Brazil (latitude 23°23′ S, longitude 51°11′ W and elevation of 566 m). Herbaceous cuttings from the median part of shoots of 'Powderblue' (*Vaccinium* sp.) blueberry stock plants were used and maintained in the same department in a greenhouse, originating from a collection of blueberry cultivars belonging to the Brazilian Agricultural Research Corporation, Pelotas, RS, Brazil.

### 2.2. Experimental Design and Treatments Description

The experimental design was completely randomized, with six treatments and four replicates, for a total of 24 experimental units, with each plot comprised of 10 cuttings. The effect of different concentrations of IBA in talc and the association with *A. brasilense* was assessed. The following treatments were tested: 0 (control); 500 mg L$^{-1}$ of IBA; 1000 mg L$^{-1}$ of IBA; *A. brasilense*; 500 mg L$^{-1}$ of IBA + *A. brasilense*; and 1000 mg L$^{-1}$ of IBA + *A. brasilense*. The maximum tested dose of IBA (1000 mg L$^{-1}$) was based on Fischer et al. [19].

Before collecting the cuttings, the IBA solutions were prepared. For this, 0.05 and 0.1 g of concentrated IBA (99.9% purity; Sigma-Aldrich®, St. Louis, MO, USA) were weighed using a semi-analytical balance and dissolved in 50 mL of 100% ethanol. When the IBA was totally dissolved, the volume was adjusted to 100 mL with distilled water, and solutions with concentrations of 500 and 1000 mg L$^{-1}$ of IBA were obtained. To prepare the IBA in talc, 0.1 g of IBA was mixed in industrial inert talc (Quimidrol®, Joinville, Brazil) for a total of 100 g. For better homogenization, sufficient IBA with ethanol solutions was added to form a paste, and then transferred to an oven at 40 °C, where it remained until complete evaporation of the solvent.

Cuttings were prepared using a bevel cut just below a node, discarding the leaves at the basal part, and keeping two pairs of leaves in the upper part which were later cut in half. During preparation, the cuttings were kept in a container with water to avoid dehydration. After preparing the cuttings, the IBA talc was applied to the base of the cuttings.

Then, the cuttings were placed in perforated plastic boxes (44 × 30 × 7 cm) containing vermiculite substrate of medium granulation for rooting, followed by the application of 1 mL of solution containing *A. brasilense* (BR 11001t) isolates obtained by Silva et al. [20]. The bacterial isolates were cultured in DYGS medium (glucose 2%, peptone 1.5%, yeast extract 2%, $KH_2PO_4$ 0.5%, $MgSO_4 \cdot 7H_2O$ 0.5%, glutamic acid 1.5%, pH 6.8), the procedure recommended by Kuss [21], with agitation of 120× *g* at

28 °C, until reaching a population density of $10^8$ CFU mL$^{-1}$. The cultured bacteria were applied directly into the substrate.

### 2.3. Growth Conditions

The boxes were arranged in a mist chamber with an intermittent misting regimen controlled by a timer and solenoid valve. The valve was programmed to mist for 10 s every 6 min. The misting nozzle employed (Model DAN-7755 Modular Greenhouse Sprinkler, Tel Aviv, Israel) had a flow rate of 35 L h$^{-1}$. The mist chamber was placed in a greenhouse with a transparent polyethylene film and 30% cover.

To control fungal diseases, the cuttings were treated weekly with a spray of tebuconazole fungicides (1 mL L$^{-1}$). Foliar fertilization with Biofert Plus® (Contagem, Brazil) fertilizer (8-9-9 + micronutrients) at a 5 mL L$^{-1}$ concentration per spray was applied every 15 d.

### 2.4. Evaluations

After 90 d, the following variables were assessed: rooted cuttings (% of cuttings that developed at least one root); survival of cuttings (% of live cuttings); foliar retention (% of cuttings that did not lose leaves); sprouting (shoot growth); cuttings with callus (% of live cuttings without roots); root dry mass per cutting (g); number of roots per cutting; and root length (cm). The root dry mass was determined by oven drying samples with forced air circulation at 78 °C for 24 h.

### 2.5. Statistical Analysis

Prior to individual analysis of variance, the homoscedasticity and normality of the residues were confirmed using Bartlett [22] ($p \leq 0.05$) and Lilliefors [23] tests ($p \leq 0.05$), respectively. Arc-sine transformations $\sqrt{(x/100)}$ were performed for variables expressed in percentage and the transformation $\sqrt{(x + 1)}$ was performed for counting data. After performing the individual analysis of variance, those characteristics which had fewer than seven ratios, between the highest and the lowest residuals, were submitted to factorial analysis of variance. Means of significantly affected characteristics were compared by Tukey's test ($p \leq 0.05$).

## 3. Results and Discussion

There were no significant differences for % rooted cuttings, survival, foliar retention, cuttings with callus, and root dry mass (Table 1). In contrast to our results, olive-tree cuttings (*Olea europaea*) treated with 3 g L$^{-1}$ of IBA in association with *A. brasilense* led to a higher percentage of rooted cuttings [24]. Therefore, it is likely that the application of IBA and the inoculation with rhizobacteria present different responses depending on the species.

**Table 1.** Rooted cuttings (RC), survival (SV), foliar retention (FR), leaf retention (LR), sprouting (S), cuttings with callus (CC), root dry mass (DM), number of roots (NR), and root length (RL) of 'Powderblue' blueberry cuttings exposed to indolebutyric acid and *Azospirillum brasilense*.

| Sources of Variation | RC (%) | SV (%) | FR (%) | S (%) | CC (%) | DM (g) | NR | RL (cm) |
|---|---|---|---|---|---|---|---|---|
| TMS [z] | 0.1 [ns] [y] | 0.3 [ns] | 0.4 [ns] | 0.8 * | 0.4 [ns] | 5.5 [ns] | 27.3 * | 66.5 * |
| EMS | 0.3 | 0.2 | 0.4 | 0.2 | 0.3 | 4.1 | 8.5 | 11.4 |
| Mean [x] | 45.4 | 89.2 | 42.0 | 73.3 | 37.1 | 0.6 | 3.9 | 4.5 |
| CV (%) | 16.7 | 8.8 | 12.3 | 12.2 | 20.8 | 33.8 | 15.0 | 26.6 |

[z] TMS = Treatment mean square. EMS = Error mean square. [y] * = significant $p \leq 0.05$ by ANOVA. ns: not significant. [x] Mean across treatments. CV (%) = Coefficient of variation.

The applications of IBA and the rhizobacterium *A. brasilense* increased the number of roots per cutting (Tables 1 and 2). One of the great advantages of increased lateral root production through

cuttings is that the volume of soil available to the plant is higher, indirectly increasing the capture of available resources in the soil and increasing the absorption of water and nutrients. These results were also observed with 'Berkeley' blueberry from the application of 1000 mg L$^{-1}$ IBA [25] and with 'Bluegem' and 'Powderblue' blueberries from the application of 2000 mg L$^{-1}$ IBA [8].

**Table 2.** Number of roots (NR), root length (RL), and sprouting (S) of 'Powderblue' blueberry cuttings treated with indolebutyric acid (IBA) and *Azospirillum brasilense* (Azo).

| Treatments | NR [1] | RL (cm) [1] | S (%) [1] |
|---|---|---|---|
| Control | 2.79 c | 2.81 b | 92.5 a |
| 500 mg L$^{-1}$ IBA | 4.02 b | 5.62 a | 85.0 a |
| 1000 mg L$^{-1}$ IBA | 3.50 b | 5.18 a | 60.0 b |
| Azo | 5.57 a | 3.88 ab | 70.0 b |
| Azo + 500 mg L$^{-1}$ IBA | 4.38 b | 4.28 ab | 75.0 b |
| Azo + 1000 mg L$^{-1}$ IBA | 3.38 b | 5.00 a | 57.5 c |

[1] Means followed by the same letter in a column do not differ by the Tukey's test set at 5% probability.

In addition, it was possible to observe that inoculation of *A. brasilense* alone was superior to the other treatments in increasing the number of roots per cutting. *A. brasilense* is known for the ability to alter the architecture of the roots of plants, increasing the formation of lateral and adventitious roots and root hairs. This is because of the production or metabolization by *A. brasilense* of chemical signaling compounds that alter root elongation and arrangement and the formation of root hairs [26].

However, there were no significant differences between treatments with IBA alone and in association with *A. brasilense*. Therefore, such an association did not benefit the development of new roots compared to the treatments with IBA alone. Indolacetic acid (IAA) is one of the most important molecules produced by *Azospirillum* sp. and studies suggest bacterial biosynthesis of IAA can be drastically affected by environmental conditions and exposure to other compounds [16,18]. Thus, the application of IBA may have influenced the production of the rhizobacteria growth-promoting substances.

The mean root length (Table 2) increased with applications of 500 and 1000 mg L$^{-1}$ of IBA and 1000 mg L$^{-1}$ of IBA + *A. brasilense,* and cuttings showed more developed roots compared to that of the control. Genetic factors can influence cell differentiation and root formation in cuttings, and different blueberry cultivars show distinct responses regarding the rooting of cuttings. In studies conducted with 'Climax' and 'Florida' blueberries, the application of IBA did not increase root length of the roots [27], but 'Bluegem', 'Bluebelle', and 'Powderblue' blueberries responded exponentially to the different concentrations of the growth regulator [8].

Regarding sprouting, emergence of growing buds was more vigorous in the control as compared to treatment with 500 mg L$^{-1}$ of IBA. Initiation and development of buds are independent of the adventitious formation of roots. The application of high concentrations of auxins in cuttings can inhibit bud growth, sometimes even preventing growth of the aerial part, although root formation may be adequate [11]. Therefore, it was verified that application of *A. brasilense* alone is a viable alternative to increasing the number of adventitious roots in 'Powderblue' blueberry cuttings.

## 4. Conclusions

The application of IBA with the *Azospirillum brasilense* rhizobacteria increased the number of roots of 'Powderblue' blueberry cuttings, while the treatments with IBA alone and IBA 1000 mg L$^{-1}$ + *A. brasilense* increased the root length of cuttings. However, treatments with IBA and *A. brasilense* combined had no impact on % rooted cuttings and % survival of cuttings.

**Author Contributions:** R.K. and D.M.Z. conceived and designed the experiments. R.K. and H.M.S. performed the experiments. R.K., W.A.R.J., R.T.F., L.S.A.G. and S.R.R. wrote the manuscript. D.M.Z. analyzed the data.

**Funding:** This study was funded by the Brazilian Council for Scientific and Technological Development (CNPq) and Coordination for the Improvement of Higher Education Personnel (CAPES).

**Conflicts of Interest:** The authors declare no conflicts of interest.

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
