# Peer review of "Association of Indolebutyric Acid with Azospirillum brasilense in the Rooting of Herbaceous Blueberry Cuttings"

_horticulturae, doi:10.3390/horticulturae5040068_

Round 1

Reviewer 1 Report

This work looks at the influence of auxins and soil bacteria on root proliferation in blueberry cuttings. The protocols developed have application to propagation of the plant through cuttings in a plant nursery setting. Six treatments involving 3 levels of IBA and the presence or absence of rhizobacteria were examined. The authors note 4 replicates were performed. Responses measured included % rooted, % survival, foliar retention, roots per cutting, root length, root dry mass, callus presence, and bud presence.

The authors note that ANOVA was performed on transformed data. Root numbers and root length increased relative to control, but the sprouting yield was adversely affected by the treatments. Other response measured showed no changes due to treatment for the chosen p value.

The work demonstrates increased rooting due to the presence of auxin and rhizobacteria, compared to their absence. IBA, however, does not statistically influence rooting outcomes when associated with bacteria treatments. This result suggests the protocol does not require IBA as a factor promoting increased viability (using the authors choice of responses) of blueberry cuttings. The authors are clear to point this out and the compact presentation of the manuscript reinforces the "neutral" result.

The manuscript could benefit from an increased explanation of the experimental design, that is currently captured in only 2 sentences (lines 103-105) and an additional comment in lines 67-69. It is clear that a 2-factor, mixed level design is proposed, with 4 replicates; this suggests 24 experiments? The authors note, however, that each plot contained 10 cuttings - were there 24 plots used with plots as the experimental unit? Since 24 and 10 are not multiples of each other, this needs to be made clear.

Author Response

First of all, we would like to thank the reviewer for the suggestions and comments. Below are our responses:

The manuscript could benefit from an increased explanation of the experimental design, that is currently captured in only 2 sentences (lines 103-105) and an additional comment in lines 67-69. It is clear that a 2-factor, mixed level design is proposed, with 4 replicates; this suggests 24 experiments? The authors note, however, that each plot contained 10 cuttings - were there 24 plots used with plots as the experimental unit? Since 24 and 10 are not multiples of each other, this needs to be made clear. The trial consisted of 6 treatments and 4 repetitions (24 plots), and each plot considered na experimental unit containing 10 cuttings. The text has been altered and the information was added.

Reviewer 2 Report

The manuscript contributes to research a treatment for the rooting of blueberry cuttings for asexual propagation, using the phytohormone indolebutyric acid and the proteobacteria Azospirillum brasilense. Overall, the manuscript sounds informative and although the topic fits in the journal Horticulturae, the authors have not clearly described the different sections. There is a lack of development of the conclusions, as well as other comments listed below:

1.      The highest dose that was tested for IBA is only 1000 mg / L when in bibliographic reference 8 (where the effectiveness of rooting of the IBA in different blueberry cultivars is proven) the highest number of roots for the Powderblue variety corresponds with a dose of 2000 mg / L of IBA. Would not you consider that better results would be obtained by testing a dose of 2000 mg / L of IBA?

2.      Table 2 shows the Tukey's test for the parameters observed in this study, while Table 1 shows the data in detail of those parameters that have given significant differences. It is for this reason that it would be better to change the order of appearance of the two tables, properly reorganizing the text.

3.      It is convenient to separate material and methods in different subsections explaining each procedure in detail.

4.      It would be very interesting to specify the conditions, such as temperature ranges, at which herbaceous cuts were maintained in the greenhouse (line 63). Since as it was concluded in the referenced work, the heating of the substrate positively affected the rooting.

5.      The asterisk in Table 2 seems to refer to Tukey's test was set at 1% and 5%. However, both material and methods (line 104) and results and discussion (line 142) appears reflected was Only 5%. This aspect must be corrected.

6.      Bibliographic articles number 15 and 22 are not mentioned in the manuscript. On the other hand, the most relevant articles referenced for the work are not in English, it would be convenient to support the work with articles in English.

-          Line 34: Change [2, 3, 4] to [2-4].

-          Line 53: Change MgSO4.7H2O to MgSO4·7H2O.

-          Line 54: Change [16, 17, 18] to [16-18].

-          Line 107: Change Discusiom to Discussion.   

-          Line 152 and 153: Delete (IBA) and (AZO).

-          Line 153: What does "Mean quare" refer to at the top of the table?

Author Response

First of all, we would like to thank the reviewer for the comments and suggestions. Below are our responses:

The highest dose that was tested for IBA is only 1000 mg / L when in bibliographic reference 8 (where the effectiveness of rooting of the IBA in different blueberry cultivars is proven) the highest number of roots for the Powderblue variety corresponds with a dose of 2000 mg / L of IBA. Would not you consider that better results would be obtained by testing a dose of 2000 mg / L of IBA? Based on trials carried out by Fisher et al. (2008) (see below), the application of 1000 mg/L of IBA increased the cutting rooting of ‘Powderblue’ blueberry in 17.5% compared with control, however, the application of 2000 mg/L of IBA reduced the rotting by 20%. Thus, the application of 1000 mg/L combined to the bacteria could increase the cutting rooting with no toxic effect.

Fischer, Doralice Lobato de Oliveira, Fachinello, José Carlos, Antunes, Luís Eduardo Corrêa, Tomaz, Zeni Fonseca Pinto, & Giacobbo, Clevison Luiz. (2008). Efeito do ácido indolbutírico e da cultivar no enraizamento de estacas lenhosas de mirtilo. Revista Brasileira de Fruticultura, 30(2), 285-289. https://dx.doi.org/10.1590/S0100-29452008000200003

shows the Tukey's test for the parameters observed in this study, while Table 1 shows the data in detail of those parameters that have given significant differences. It is for this reason that it would be better to change the order of appearance of the two tables, properly reorganizing the text. Changes made.

It is convenient to separate material and methods in different subsections explaining each procedure in detail. Changes made.

It would be very interesting to specify the conditions, such as temperature ranges, at which herbaceous cuts were maintained in the greenhouse (line 63). Since as it was concluded in the referenced work, the heating of the substrate positively affected the rooting.

Reviewer 3 Report

The authors have presented a manuscript detailing the impact of co-application of synthetic auxin (IBA) and rhizobacterium (Azospirillum) on the growth of blueberry cuttings. In general the authors have done a good job of presenting the experimental results. Data are clearly and concisely presented, and the conclusions are supported by the data shown. Experimental protocols are, for the most part, clear and replicatable. However, some issues could be more clearly addressed by the authors:

Introduction:

In lines 48-49, the text would be improved by reading “Azospirillum is a genus of PGPR”

Methods:

The specific alcohol used to solubilize IBA should be clarified. Did the authors use ethanol, methanol, or another alcohol?

IBA pastes were kept at 40C. While not as unstable as IAA, IBA does exhibit some degree of thermal instability. This should be noted by the authors, as it means that treatments of plants will likely be at concentrations lower than those calculated by the authors.

Discussion and conclusions:

The authors have done a good job of presenting the primary conclusion regarding root number, that the number of roots was most increased by Azo treatments. The authors have done a good job in noting that Azo +IBA treatments are similar to IBA in terms of number of roots.

However, the authors could more clearly present the root length data. The authors present the treatments impacting root length and note that IBA increased RL the most. However, it should be stated more definitively that treatments with both Azo and IBA are not different from treatments with IBA alone.

The authors conclude that application of IBA may be impacting the production of growth-promoting compounds by Azospirillum. This is a good conclusion, but ignores other potential explanations of the observed phenotypes. For example, IBA treatments could be altering auxin uptake and/or metabolism in the blueberry cuttings themselves. Additionally, application of IBA treatments could be altering the pH of the media, impacting either the secretion of bacterially produced compounds, or the uptake of these compounds by blueberry cuttings.

Author Response

First of all, we would like to thank the reviewer for the suggestions and comments. Below are our responses:

In lines 48-49, the text would be improved by reading “Azospirillum is a genus of PGPR” Changes made.

The specific alcohol used to solubilize IBA should be clarified. Did the authors use ethanol, methanol, or another alcohol? The ethanol 100% has been used. Changes made.

The authors have done a good job of presenting the primary conclusion regarding root number, that the number of roots was most increased by Azo treatments. The authors have done a good job in noting that Azo +IBA treatments are similar to IBA in terms of number of roots. However, the authors could more clearly present the root length data. The authors present the treatments impacting root length and note that IBA increased RL the most. However, it should be stated more definitively that treatments with both Azo and IBA are not different from treatments with IBA alone. We agree that the treatments with the application of Azo and IBA are not diferente of IBA alone. However, these treatments are not different of control, what could be make some confusion in Discussion section.

The authors conclude that application of IBA may be impacting the production of growth-promoting compounds by Azospirillum. This is a good conclusion, but ignores other potential explanations of the observed phenotypes. For example, IBA treatments could be altering auxin uptake and/or metabolism in the blueberry cuttings themselves. Additionally, application of IBA treatments could be altering the pH of the media, impacting either the secretion of bacterially produced compounds, or the uptake of these compounds by blueberry cuttings. We thank the contributions and suggestions made by the reviewers. However, the pH of the medium has been assessed in this trial, and for this reason, it is not possible to add it in the Discussion section, but it will be considered in future trials.

Round 2

Reviewer 2 Report

The article has improved the vast majority of the points of the review, thus improving the quality of the article. It would be convenient to include the article referred to (Fisher et al., 2008) so that readers know the reason for testing only up to 1000 mg / L.

Despite the corrections made by the authors there are two important aspects that have not been corrected. One of them is a further development of the conclusions. The other is a greater use of bibliography in English, due to the difficulties that this entails for readers of international journals to collate or deepen in the study.

Author Response

Dear Reviewer, first of all, thank you very much for your contributions. The changes were made according to your comments.

Below you find our specific responses:

1 - The article has improved the vast majority of the points of the review, thus improving the quality of the article. It would be convenient to include the article referred to (Fisher et al., 2008) so that readers know the reason for testing only up to 1000 mg / L.

Answer: The citation Fisher et al. (2008) was added to the text in Material and Methods section.

2- Despite the corrections made by the authors there are two important aspects that have not been corrected. One of them is a further development of the conclusions. The other is a greater use of bibliography in English, due to the difficulties that this entails for readers of international journals to collate or deepen in the study.

Answer: The development of conclusions were changed (see in Abstract and in Conclusions sections). Indeed, the most of the bibliography used in this article is in English, and only two of them are in a different language (citations#5 and 9). Actually, most of the Journals demand English as the official language for publication, so that is the reason of the greater use of bibliography in English in our manuscript. We believe that readers of international journals have several tools to collate or deepen in the study, and keeping most of the citations in English would be better for most of the readers.
